# Genotyping by Sequencing Reads of 20 *Vicia faba* Lines with High and Low Vicine and Convicine Content

**Felix Heinrich [1]** , **Mehmet Gültas [1,2]** , **Wolfgang Link [3] and Armin Otto Schmitt [1,2],\***

[1] Breeding Informatics Group, Department of Animal Sciences, Georg-August University, Margarethe von Wrangell-Weg 7, 37075 Göttingen, Germany; felix.heinrich@uni-goettingen.de (F.H.); gueltas@cs.uni-goettingen.de (M.G.)

[2] Center for Integrated Breeding Research (CiBreed), Albrecht-Thaer-Weg 3, Georg-August University, 37075 Göttingen, Germany

[3] Department of Crop Sciences, Georg-August University, Von-Siebold-Str. 8, 37075 Göttingen, Germany; wlink@gwdg.de

\* Correspondence: armin.schmitt@uni-goettingen.de

**Abstract:** The grain faba bean (*Vicia faba*) which belongs to the family of the Leguminosae, is a crop that is grown worldwide for consumption by humans and livestock. Despite being a rich source of plant-based protein and various agro-ecological advantages its usage is limited due to its anti-nutrients in the form of the seed-compounds vicine and convicine (V+C). While markers for a low V+C content exist the underlying pathway and the responsible genes have remained unknown for a long time and only recently a possible pathway and enzyme were found. Genetic research into *Vicia faba* is difficult due to the lack of a reference genome and the near exclusivity of V+C to the species. Here, we present sequence reads obtained through genotyping-by-sequencing of 20 *Vicia faba* lines with varying V+C contents. For each line, ~3 million 150 bp paired end reads are available. This data can be useful in the genomic research of *Vicia faba* in general and its V+C content in particular.

**Dataset:** The reads have been submitted to the European Nucleotide Archive (ENA) under the accession PRJEB38838.

**Keywords:** *Vicia faba*; GBS; vicine/convicine

---

## 1. Summary

The Protein Crop Strategy of the German Federal Ministry of Food and Agriculture has the goal of raising the importance of domestic protein crops e.g., legumes in Germany and Europe in order to improve ecosystem services and resource conservation as well as to reduce the dependency on imported crops [1]. The faba bean (*Vicia faba*) is a prime candidate for this strategy being a globally grown legume that has several agro-ecological advantages (N-symbiosis, rotation hygiene, and pollinator support) and serving as food for humans and livestock [2]. Regardless of these benefits its usage is limited due to the anti-nutrients vicine and convicine (V+C) that occur in their seeds. These compounds have negative effects to animals as well as to humans suffering from G6PD deficiency [3,4]. Despite ongoing research efforts and the discovery of a robust marker for the V+C content, the responsible genes and mechanisms remained unknown for a long time and the location of

the locus could only be restricted to an interval on chromosome 1 of *Vicia faba* that shows conserved synteny with a region on chromosome 2 of the related species *Medicago truncatula* that is about 900,000 bp long [3]. Research has been exacerbated due to the lack of an annotated reference genome for *Vicia faba*, which is assumed to be about 13 Gbp [5]. In a recent preprint the authors have found an enzyme associated with V+C biosynthesis and identified it as a guanosine triphosphate (GTP) cyclohydrolase II, proposing the purine GTP as a precursor for vicine [6]. The breeding of novel low V+C varieties to improve the usage of *Vicia faba* as feed is the goal of the project Abo-Vici, which is supported by the German Federal Ministry of Food and Agriculture [7]. As part of this project we obtained reads from 20 *Vicia faba* lines with known V+C content through genotyping-by-sequencing (GBS). We offer this data here for the benefit of researches into *Vicia faba* and its V+C content in particular. We have so far successfully used this data for the prediction of regulatory regions in *Vicia faba* and the identification of regulatory single nucleotide polymorphisms (SNPs) that are associated with V+C content [8]. For this we built a partial genome for *Vicia faba* from the GBS reads that spanned ~1 % of the total genome and performed variant calling with it, which resulted in more than 600,000 high quality SNPs. This partial genome is available upon request from the corresponding author. Finally, while the data itself are not enough alone, the data can support the eventual creation of an annotated reference genome for *Vicia faba*.

## 2. Data Description

The sequence reads for 20 *Vicia faba* lines obtained through GBS are stored as paired end reads in two FASTQ files per *Vicia faba* line. These FASTQ files contain both the nucleotide sequence and its corresponding quality scores as text. Per sample ~3 million 150 bp paired end reads are available, such that the total amount of sequence amounted to 18 Gbp. The uncompressed data required 51 GB of disk space. The sequences have been deposited at the European Nucleotide Archive (ENA) under the accession number PRJEB38838.

## 3. Methods

*Plant Material and Sequencing*

We obtained GBS data from 20 inbred lines of *Vicia faba*. The lines were inbred via single-seed descent from cultivars, from a gene-bank accession, from biparental crosses or from a landrace and include winter and spring types (see Table 1 for more information). Six of the lines had a low V+C content and 14 had high V+C content. DNA extraction, sequencing and filtering were carried out by LGC Genomics GmbH (Berlin, Germany). The DNA was extracted via LGC's sbeadex livestock kit following the lysis protocol L for plant tissue from the grains of the plants. From each line two pooled grains were used. An extraction using the sbeadex plant kit was tested but provided poorer results than the livestock kit. For each library construction 100–200 ng of genomic DNA were used, which was quantified with a NanoDrop. The DNA was digested with 2 units of the restriction enzyme MslI (NEB, recognition sequence: CAYNN^NNRTG) in NEB4 buffer in 20 µL volume for 2 h at 37 °C. The restriction enzyme was heat inactivated by incubation at 80 °C for 20 min. The TrueSeq adapter sequences used were:

- adapter_prefix_R1 'AGATCGGAAGAGCGGTTCAGCAGGAATGCCGAGACCGATC'
- adapter_prefix_R2 'AGATCGGAAGAGCGTCGTGTAGGGAAAGAGTGTAG'

For the ligation 10 µL of each restriction digest were transferred to a new 96well PCR plate, mixed on ice first with 1.5 µL of one of 96 inline-barcoded forward blunt adaptors (pre-hybridized, concentration 5 pmol/µL), followed by addition of 20 µL Ligation master mix (contains: 15 µL NEB Quick ligation buffer, 0.4 µL NEB Quick Ligase, 7.5 pmol pre-hybridized common reverse blunt adaptor). Ligation reactions were incubated for 1 h at room temperature, followed by heat inactivation for 10 min at 65 °C. After the ligation reactions the libraries were purified using

Agencourt XP beads. Following that the libraries were size selected by a size selection on a LMP-agarose gel, removing fragments smaller than 300 bp or larger than 400 bp. For the final quality control, Fragment Analyzer and Qubit were used. The libraries were then amplified in 20 µL PCR reactions using MyTaq (Bioline) and standard Illumina TrueSeq amplification primers. The number of cycles was limited to 14. Having each line uniquely barcoded the samples were pooled and ran on the same sequencing run. An Illumina NextSeq 500 V2 platform was then used for genotyping-by-sequencing. Demultiplexing of the libraries was done using the Illumina `bcl2fastq 2.17.1.14` software. Finally sequencing adapter remnants were clipped using `cutadapt 1.13+18` [9] and reads whose 5′ ends did not match the restriction enzyme site were discarded. As a last step FastQC reports (https://www.bioinformatics.babraham.ac.uk/projects/fastqc/) were generated for the FASTQ files.

**Table 1.** Vicine and convicine status of the 20 lines, name, sample ID, additional notes and European Nucleotide Archive (ENA) accession number under which the sample data is available.

| ENA Accession | Sample ID | V+C | Line | Notes |
|---|---|---|---|---|
| ERS4652931 | Sample_8 | Low | Line 1268-4-1 | Ancestor of low V+C content |
| ERS4652926 | Sample_3 | Low | Mélodie/2 | cv. Mélodie; minor, spring bean |
| ERS4652927 | Sample_4 | Low | F7(Mélodie/2 x ILB938/2)-139-1-1 | Near isogenic lines (ILB938/2 is from Ecuador) |
| ERS4652928 | Sample_5 | Low | F7(Mélodie/2 x ILB938/2)-201-3-1 | |
| ERS4652932 | Sample_9 | High | F7(Mélodie/2 x ILB938/2)-139-2-1 | |
| ERS4652933 | Sample_10 | High | F7(Mélodie/2 x ILB938/2)-201-4-1 | |
| ERS4652929 | Sample_6 | Low | F7[VC.14.8099-843-2-1] | Near isogenic lines from a breeder's cross, spring beans |
| ERS4652930 | Sample_7 | Low | F7[VC.14.8099-848-3-1] | |
| ERS4652934 | Sample_11 | High | F7[VC.14.8099-843-3-3] | |
| ERS4652935 | Sample_12 | High | F7[VC.14.8099-848-4-1] | |
| ERS4652924 | Sample_1 | High | HediLin-1 | cv. Hedin; minor, spring bean |
| ERS4652936 | Sample_13 | High | PietraLin | Major, Mediterranean bean |
| ERS4652937 | Sample_14 | High | (HediLin/1 x PietraLin)-2-4 | Near isogenic lines |
| ERS4652938 | Sample_15 | High | (HediLin/1 x PietraLin)-4-4 | |
| ERS4652939 | Sample_16 | High | S_281 | Academic winter bean lines |
| ERS4652940 | Sample_17 | High | S_301 | |
| ERS4652941 | Sample_18 | High | S_034 | |
| ERS4652942 | Sample_19 | High | S_290 | |
| ERS4652925 | Sample_2 | High | Hiverna/2 | cv. Hiverna; minor, winter bean |
| ERS4652943 | Sample_20 | High | Côte d'Or/1 | Côte d'Or; minor, winter bean |

**Author Contributions:** M.G. designed and supervised the research. F.H. participated in the design of the study, prepared the data sets and conducted the bioinformatics analysis together with M.G. A.O.S. and W.L. secured the funding for data acquisition. W.L. provided seed of the inbred lines and expertise with the crop plant. F.H., M.G. and A.O.S. wrote the final version of the manuscript. M.G. and A.O.S. supervised the writing of the manuscript, conceived as well as managed the project. All authors have read and agreed to the published version of the manuscript.

**Funding:** This research was partially funded by the Lower Saxony Ministry of Science and Culture, grant number MWK 11-76251-99-30/16.

**Acknowledgments:** We acknowledge support by the German Research Foundation and the Open Access Publication Funds of the University of Göttingen.

**Conflicts of Interest:** The authors declare that the research was conducted in the absence of any commercial or financial relationships that could be construed as a potential conflict of interest.

## Abbreviations

The following abbreviations are used in this manuscript:

GBS    Genotyping by Sequencing
SNP    Single Nucleotide Polymorphism
V+C    Vicine and Convicine

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
