# Peer review of "Genotyping by Sequencing Reads of 20 Vicia faba Lines with High and Low Vicine and Convicine Content"

_data_

Round 1
Reviewer 1 Report
The data presented in this note seem to be very straight forward. As presented, the methods are repeatable and were completed as expected. It would be a nice addition to have the number of PCR cycles included for the library preparation. This could give the readers a better sense if these were low- to no-PCR reactions or standard.
Line 48: "6" should be written out as "Six".
Lines 49: Perhaps write out the full name of the company.
Author Response
We would like to thank the reviewer for these thoughtful comments and efforts towards improving our manuscript. The manuscript was revised and modified accordingly. The manuscript was completed by new text, which was highlighted in red color.
It would be a nice addition to have the number of PCR cycles included for the library preparation. This could give the readers a better sense if these were low- to no-PCR reactions or standard.
We have added the number of PCR cycles, which was 14.
Line 48: "6" should be written out as "Six".
It was corrected.
Line 49: Perhaps write out the full name of the company.
We mention the full name of the company in line 54 (LGC Genomics GmbH).
Reviewer 2 Report
Heinrich et al. describe a GBS dataset for 20 faba bean lines that differ in vicine and convicine content. Overall, the paper is well written and the dataset will be a nice resource for faba bean research. Below I have outlined several suggestions. In general, slightly more details are needed for the methodology, particularly regarding the library preparation.
Data description:
- L40-43: I suggest the authors double-check the math on the description of total amount of sequence. If the 3 million reads per sample refers to pairs of reads, then I calculate 18 Gb of total sequence (20 samples x 3,000,000 reads x 150 bp x 2 reads per pair)
Methodology:
- How much input DNA was used for each library construction? How was DNA quantified?
- Details are needed for the digestion reactions. At what temperature and how long did the reactions proceed? What reagents did the reaction consist of?
- What are the adapter sequences? Similarly, more details are needed for the ligation. What temperature, length of incubation, contents of reaction, etc.?
- Were the libraries purified after digestion/ligation reactions?
- Was each line uniquely barcoded? Were the samples pooled and ran on the same sequencing run? These details should be added into the manuscript.
- Were the libraries amplified before pooling?
- Digestion will result in many fragments of a multitude of sizes. Were the libraries size-selected in some manner? If so, how was that completed?
- Methods seems out of logical order. For example, L49-51 talk about DNA extraction. L52 mentions the sequencing. Then the following sentence mentions the digestions/library pep. I suggest reordering the section in a logical manner.
- L54-56 mention some post-sequencing quality control, such as trimming adapter sequences. However, the software used to process these reads was not mentioned.
Table 1: I suggest adding the ENA accession number for each line in the table. This would make it easier to search.
Author Response
We would like to thank the reviewer for these thoughtful comments and efforts towards improving our manuscript. The manuscript was revised and modified accordingly. The manuscript was completed by new text, which was highlighted in red color.
L40-43: I suggest the authors double-check the math on the description of total amount of sequence. If the 3 million reads per sample refers to pairs of reads, then I calculate 18 Gb of total sequence (20 samples x 3,000,000 reads x 150 bp x 2 reads per pair)
We were referrring to the total amount of data , which was about 51 GB of disk space. We include now also the amount of sequence data, which is 18 Gbp.
How much input DNA was used for each library construction? How was DNA quantified?
We have included the amount of DNA and how it was quantified.
Details are needed for the digestion reactions. At what temperature and how long did the reactions proceed? What reagents did the reaction consist of?
We have included this information.
What are the adapter sequences? Similarly, more details are needed for the ligation. What temperature, length of incubation, contents of reaction, etc.?
We have incorporated the correspoinding details into the manuscript.
Were the libraries purified after digestion/ligation reactions?
The libraries were purified and the details have been included.
Was each line uniquely barcoded? Were the samples pooled and ran on the same sequencing run? These details should be added into the manuscript.
Each Vicia faba line was uniquely barcoded and all samples ran on the same run. The details have been added.
Were the libraries amplified before pooling?
Yes, and we have added that information.
Digestion will result in many fragments of a multitude of sizes. Were the libraries size-selected in some manner? If so, how was that completed?
Yes, fragments between 300 and 400 bp habe been selected. We have added this information.
Methods seems out of logical order. For example, L49-51 talk about DNA extraction. L52 mentions the sequencing. Then the following sentence mentions the digestions/library pep. I suggest reordering the section in a logical manner.
We agree with the recommendation and have changed the order.
L54-56 mention some post-sequencing quality control, such as trimming adapter sequences. However, the software used to process these reads was not mentioned.
The used software (cutadapt) has been added.
Table 1: I suggest adding the ENA accession number for each line in the table. This would make it easier to search.
We agree with the suggestion and have included the accession numbers into the table.
Reviewer 3 Report
This is the valuable study providing important sequence reads of Vicia faba that have different V+C contents.
The study updates us with sequence reads obtained through genotyping-by-sequencing for 20 lines of Vicia faba that have different V+C contents. The data have been also submitted to the European Nucleotide Archive (ENA).
The readability of the manuscript is fluent and understandable. There are some minor concerns which should be addressed before accepting for publication:
- Additional English editing should be done; please check the use of prepositions (I am not a native English speaker but there might be some corrections needed for the final tuning of the manuscript).
- Key words should not be repeated if already stated in the title. Please check and correct.
- The abstract is to general. Please include key information from section “Data description”.
- Please describe more in detail the methods which have been applied by LGC including bioinformatics software for QC of Fastq files,…
- Please check the references (style and correct citation).
Author Response
We would like to thank the reviewer for these thoughtful comments and efforts towards improving our manuscript. The manuscript was revised and modified accordingly. The manuscript was completed by new text, which was highlighted in red color.
Additional English editing should be done; please check the use of prepositions (I am not a native English speaker but there might be some corrections needed for the final tuning of the manuscript).
We agree with this remark of our reviewer and our colleagues controlled the manuscript again. To the best of our knowledge, we corrected and rephrased the corresponding sentences, wherever it seemed necessary, in the revised version of our manuscript. For the sake of readability, we did not mark these minor changes in the text.
Key words should not be repeated if already stated in the title. Please check and correct.
This change would remove all keywords. Since we could not find such a recommendation in the author instructions we leave the decision to remove the keywords to the editors.
The abstract is to general. Please include key information from section “Data description”.
We have included the information (see L9-10).
Please describe more in detail the methods which have been applied by LGC including bioinformatics software for QC of Fastq files,…
We have expanded the method section substantially and have included these details.
Please check the references (style and correct citation).
We have made sure that the references are in the style which is appropriate for Data and made some minor amendments.
Revisions regarding the additional comment:
Please correct the statement about vicine-convocine pathway in the abstract "underlying pathway and the responsible genes are to date still unknown". This statement is incorrect as the VC pathway has been pre-published at https://www.biorxiv.org/content/10.1101/2020.02.26.966523v1.abstract
We have changed the statement and added a reference to the article (see L4-6 and L30-32).